# Virus-like Particle (VLP) Vaccines for Cancer Immunotherapy

**DOI:** 10.3390/ijms241612963

**Published:** 2023-08-19

**Authors:** Francesca Ruzzi, Maria Sofia Semprini, Laura Scalambra, Arianna Palladini, Stefania Angelicola, Chiara Cappello, Olga Maria Pittino, Patrizia Nanni, Pier-Luigi Lollini

**Affiliations:** 1Department of Medical and Surgical Sciences (DIMEC) and Alma Mater Institute on Healthy Planet, University of Bologna, 40126 Bologna, Italy; francesca.ruzzi2@unibo.it (F.R.); mariasofia.semprini2@unibo.it (M.S.S.); laura.scalambra2@unibo.it (L.S.); stefania.angelicola2@unibo.it (S.A.); chiara.cappello4@unibo.it (C.C.); olgamaria.pittino@studio.unibo.it (O.M.P.); patrizia.nanni@unibo.it (P.N.); 2Department of Molecular Medicine, University of Pavia, 27100 Pavia, Italy; arianna.palladini@unipv.it

**Keywords:** cancer immunotherapy, cancer immunoprevention, cancer vaccines, virus-like particles (VLPs), tumor antigens

## Abstract

Cancer vaccines are increasingly being studied as a possible strategy to prevent and treat cancers. While several prophylactic vaccines for virus-caused cancers are approved and efficiently used worldwide, the development of therapeutic cancer vaccines needs to be further implemented. Virus-like particles (VLPs) are self-assembled protein structures that mimic native viruses or bacteriophages but lack the replicative material. VLP platforms are designed to display single or multiple antigens with a high-density pattern, which can trigger both cellular and humoral responses. The aim of this review is to provide a comprehensive overview of preventive VLP-based vaccines currently approved worldwide against HBV and HPV infections or under evaluation to prevent virus-caused cancers. Furthermore, preclinical and early clinical data on prophylactic and therapeutic VLP-based cancer vaccines were summarized with a focus on HER-2-positive breast cancer.

## 1. Introduction

Immunization has been practiced for hundreds of years starting with the discovery of the first vaccine to prevent smallpox infection [1]. In the early 20th century, immunotherapy, as a method of treating cancer, started with the pioneering work of William Coley [2]. Since then, immunotherapy has revolutionized the treatment of cancers, mainly owing to “passive” strategies, such as monoclonal antibodies [3,4]. Cancer vaccines might be a promising active antitumor strategy, mainly due to their capability of harnessing both humoral and cellular responses and providing long-term protection [5,6,7,8].

Currently, there are vaccines approved worldwide to prevent malignancies caused by two viruses, the human hepatitis B virus and human papillomavirus [4], and only one therapeutic vaccine based on dendritic cells, sipuleucel-T, approved in 2012 for prostate cancer [9]. Of note, the development of a therapeutic cancer vaccine is more complicated than a prophylactic one, mostly because tumors, in order to progress, have already escaped the immunosurveillance [5]. Thus, a good therapeutic cancer vaccine must induce a strong T-cell response against tumor cells and elicit high and persistent antibody titers able to inhibit cancer cells [5,10].

The development of a cancer vaccine includes several crucial steps, such as the choice of the appropriate platform, target antigens, delivery system, and/or adjuvants [5,7,8,11].

Virus-like particles (VLPs) are nanoparticles that spontaneously assemble from viral structural proteins. Due to the lack of genetic material needed for viral replication, VLPs inherently ensure safety against unintentional viral gene delivery. VLPs can be used as a platform to present different classes of epitopes on their surface for various applications, in particular for vaccine development. Indeed, their ability to interact with dendritic cells (DCs) and induce a strong B cell response, as well as specific CD4 and CD8 T-cell responses, can enhance vaccine efficacy [12,13,14,15,16].

In this review, we summarize the VLP cancer vaccines successfully licensed or under development for the prevention of cancer caused by viral infections, as well as the VLP vaccines under preclinical or early clinical evaluation as therapeutic strategy for various cancer types, with a focus on HER-2-positive breast cancer models.

## 2. Virus-like Particles (VLPs): An Overview

Virus-like particles (VLPs) are complex, multimeric, self-assembled protein structures that morphologically and structurally resemble native viruses or bacteriophages but lack the viral genetic material, rendering them noninfectious and nonreplicative [17]. In essence, they are hollow shells that mimic the overall structure of a virus without causing a viral infection. According to the presence or absence of a lipid envelope, VLPs can be classified as enveloped or nonenveloped [18,19].

The first VLPs were described in the context of the hepatitis B virus in the 1960s. Baruch S. Blumberg discovered Australia antigen (later defined as hepatitis B surface antigen, HBsAg) in the serum of an Australian aborigine. He found that patients with hepatitis B infection had particles in their bloodstream that resembled the hepatitis B virus but were noninfectious. These particles, primarily composed of HBsAg proteins, lacked the viral DNA necessary for infection [20]. Since that moment, VLPs have been widely studied and engineered as drug carriers or vaccine platforms [13,14].

Producing VLPs involves several key steps, including selecting an appropriate expression system, genetic engineering of the host cell/organism, protein expression, purification, and assembly of VLPs [14,16,21,22].

Commonly used expression systems include bacteria (e.g., *Escherichia coli*) [23], yeasts (e.g., *Pichia pastoris*) [24], insect cells (e.g., *Drosophila melanogaster* S2 cells) [25,26], mammalian cells (e.g., human embryonic kidney (HEK) cells) [27,28], and even plant cells [29]. Each system has distinct characteristics that can affect both the yield of VLP production, a crucial factor considering the large quantities needed for vaccine development, and their immunogenicity [21]. For instance, bacterial and yeast systems are often favored for their rapid growth rates and high yield. They are also relatively easy to handle and can be cost-effective for large-scale production [30]. However, these systems (especially bacteria) may not correctly fold complex viral proteins or carry out glycosylation, phosphorylation, and other post-translational modifications (PTMs), impacting the structural integrity and immunogenicity of VLPs [18,21]. Due to the limited PTMs, these expression systems are generally used to generate nonenveloped VLPs [31,32]. In contrast, insect and mammalian cell systems can produce VLPs with more accurate folding and post-translational modifications, closely resembling the native virus particles. These modifications can significantly impact the ability of VLPs to trigger a strong immune response [33]. However, these systems can be more expensive and challenging to scale up [19,30].

Next, the host organism is genetically engineered to express the viral structural proteins that will form VLPs. This often involves the construction of a recombinant plasmid or virus containing the gene of interest that may be introduced in the host system. The expressed proteins are then purified from the host cells to separate the viral proteins from host proteins and other cellular components [32]. Once purified, the viral proteins self-assemble into VLPs. This process is often spontaneous, driven by the same interactions that lead to viral capsid formation in a natural viral infection [34].

Characterizing the size, morphology, composition, and stability of VLPs is crucial to understanding their properties and how they can elicit immune responses [32]. The size and morphology of VLPs play a pivotal role in how they are recognized by the immune system and the subsequent elicitation of an immune response [13,14,15,35,36].

VLPs typically range from 20 to 500 nm in diameter, similar to the size of many viruses [21,22,37]. This size factor can influence the uptake by antigen-presenting cells (APCs), such as dendritic cells, which play a key role in sparking off an immune response [38]. Particles around 20–50 nm in diameter may be directly presented to the lymph nodes and interact with B cells, while larger particles (up to 500 nm) might be primarily taken up by APCs at the injection site and then transported to the lymph nodes [22,37,38]. Thus, smaller particles have been shown to induce Th2-type responses associated with humoral immunity, while larger particles can elicit Th1-type responses, which are crucial for driving cellular immunity and eliminating intracellular pathogens and cancer cells [14,15,16,37].

Along with the size, the morphologies of VLPs, which range from icosahedral to rod-shaped and resemble pathogen-associated structural patterns (PASPs), also enhance the uptake by APC cells. Moreover, the presence of repetitive and organized structures on the surface of VLPs can enhance their immunogenicity by cross-linking B cell receptors, thereby promoting B cell activation and antibody production, resulting in high antibody titers [13,37,38]. The presence of repetitive structures on the VLP surface can also stimulate the innate immune system through the activation of pattern recognition receptors (PRRs), further enhancing the immune response [13,14,15].

Thus, VLPs have been established as multipurpose tools due to their unique structural properties, which are exploited in a variety of applications. Their internal cavity can serve as a delivery system for various payloads, such as genes, peptides, proteins, and small drugs, effectively delivering these components to target areas [32,39]. A significant benefit of VLPs is their potential for precise drug delivery, and by leveraging their characteristics of increased permeability and retention, they are especially useful for directing therapeutic agents to tumor tissues [40]. In the field of vaccines, VLPs offer an attractive solution due to their adjuvant properties linked to the size and shape features enhancing the immune response to the antigens they carry [41,42,43].

It is crucial to evaluate any possible side effects that might be related to autoimmunity caused by the antigen or allergic reactions and excessive immune responses against VLPs [14,44]. Because VLP vaccines are designed to induce both humoral and cellular response, each component (coat protein, chemical cross-linker, the displayed antigen, and adjuvant) could present potential toxicity [14]. Some examples of adverse effects are the pre-existing immunity against vaccine carrier proteins, the presence of unconjugated self-antigen that may bind healthy cells, or an excessive immune response induced by the vaccine combined with an adjuvant [14,45,46,47,48]. Thus, for every new vaccine composition and administration schedule, it is essential to study the possible toxicity. Current data from approved VLP vaccines show that they are generally safe [49].

## 3. Cancer Prevention through VLP-Based Vaccines

Viruses are known to play a role in the development of certain cancers. It is estimated that up to 20% of global cancer cases are caused by infectious agents. The main types of infection-related cancers include cervical cancer, liver cancers, stomach cancer, and some types of lymphoma [43]. Notably, human papillomavirus (HPV), hepatitis B (HBV) and hepatitis C (HCV) are among the most common virus types that lead to cancer.

The only treatments successfully applied at the population level to prevent cancers are vaccinations against cancer-causing viruses, especially HBV and HPV.

### 3.1. VLP Vaccines for the Prevention of HBV Infection and Related Cancers

The World Health Organization (WHO) estimated that 296 million people were infected with HBV in 2019, with 1.5 million new infections every year, leading to more than 800,000 deaths caused by virus-induced cirrhosis and liver cancers [50]. The development of HBV vaccines made a significant impact on public health and represents the best strategy to prevent HBV infection and HBV-induced diseases [15,51].

The first-generation HBV vaccines, licensed in 1982, were plasma-derived vaccines made from the blood serum of chronically asymptomatic infected donors [52]. Although these vaccines were effective, there were some barriers, such as public concern about the potential transmission of other blood-borne diseases, despite the rigorous purification process and the high production costs [53,54,55].

The second-generation HBV vaccines were first developed in 1986 using recombinant DNA technology. In this method, the gene encoding the HBV surface antigen (HBsAg) was cloned into yeast cells (and later in mammalian cells as third-generation vaccines), which were then cultured and induced to express the HBsAg protein [55,56,57]. These proteins self-assemble into noninfectious VLPs that mimic the virus outer shell, thus representing the first VLP-based vaccine [15,54,58]. Several vaccines against HBV have been developed and licensed over the years; most contain the small surface antigen S (HBsAgS) and are administered with an aluminum hydroxide (alum) adjuvant, such as Engerix-B (GSK), Recombivax (Merck), and PreHevbrio (SCIgen). The Heplisav-B (Dynavax) vaccine, like the other second- and third-generation ones, consists of HBsAg self-assembled VLPs but presents the CpG sequence 1018, which acts as an adjuvant and stimulates the immune system through toll-like receptor 9 (TLR-9) [59]. This vaccine induces a higher antibody response both in healthy individuals and those affected by diabetes or chronic kidney disease than Engerix-B [59,60]. A recent study showed that Heplisav-B was also effective in inducing antibody responses in patients who require chronic use of immunosuppressive drugs [61].

Although all the approved vaccines showed protective efficacy toward HBV infections, none were able to induce a complete therapeutic remission [62,63]. Zhang T et al. recently published preclinical results of a novel immuno-enhanced VLP carrier (CR-T3) derived from the round leaf bat HBV core antigen (RBHBcAg) expressing a 13-mer peptide (SEQ13) from HBsAg as therapeutic strategy for chronic hepatitis B. The CR-T3-SEQ13 vaccine induced a potent antibody response in several animal models, mediating HBsAg clearance in vivo [64].

### 3.2. VLP Vaccines for the Prevention of HPV Infection and Related Cancers

HPV is the most common sexually transmitted infection worldwide [65]. Papillomaviruses consist of more than 150 HPV types that can be divided into two main groups according to the infection site: the skin or internal squamous mucosa. Some viruses cause benign lesions, while others cause malignant cancers, particularly in the genital tract [51,66]. HPV16 and HPV18, along with viral genotypes 6, 11, 31, 33, 45, 52, and 58, are classified as high-risk types and are responsible for most HPV-related genital cancers [67].

In 1991, Jian Zhou first described the production of recombinant HPV VLPs [68]. By expressing the L1 and L2 virus capsid proteins recombinantly using vaccina virus, he observed that they self-assembled into VLPs that reproduced the original virion structure [68,69]. This study, together with many others running in parallel in several laboratories, has laid the basis for the development of HPV vaccines, the first of which was licensed by the FDA in 2006 [67,69,70,71].

Three prophylactic vaccines against HPV have been approved worldwide, all of which are based on L1 VLPs. Cervarix (GlaxoSmithKline, GSK, Brentford, UK) is a bivalent vaccine against viral genotypes 16 and 18 produced using Hi-5 baculovirus as the expression system [72,73]. Right after Cervarix, the quadrivalent vaccine Gardasil, licensed in 2006 (Merck Sharp & Dohme Corp.), was developed by harnessing yeast cells as the expression tool and used to prevent infection by viral genotypes 6, 11, 16, and 18 [74]. The Gardasil vaccine has been subsequently improved to prevent infections of nine HPV viral genotypes (6, 11, 16, 18, 31, 33, 45, 52, and 58) and approved by the FDA in 2017 as Gardasil 9 (Merck Inc., Rahway, NJ, USA).

HPV vaccines using the minor capsid protein L2 are under preclinical investigation. The L2 protein represents a promising antigen as it is involved in the penetration of HPV particles into epithelial cells and is highly conserved, possibly providing cross-protection against several HPV genotypes. Unfortunately, the L2 protein lacks the ability of self-assembling; thus, a VLP capsid multivalent display, presenting both L2 and L1 peptides or epitopes, might represent a more practical option [75,76].

Despite the excellent preventive efficacy and capability to induce neutralizing antibodies against HPV, the approved VLP vaccines for HPV are not suitable for the treatment of already infected individuals because the integration of the viral genome into the host genome leads to the loss of many early (E1, E2, E4, and E5) and late genes (L1 and L2), making prophylactic vaccines ineffective against HPV-related lesions and cancers [77]. Cancer vaccines based on live vectors, peptides, proteins, and dendritic cells mainly presenting E2, E6, or E7 are under clinical development for the treatment of precancerous lesions [78,79].

### 3.3. VLP Vaccines against Other Viruses-Causing Cancers

The excellent results achieved with prophylactic vaccines against HBV and HPV has encouraged the development of VLP vaccines for the prevention of other oncogenic viruses, such as human herpesvirus type 8 (HHV-8), also known as Kaposi sarcoma-associated herpes virus (KSHV), and the Epstein–Barr virus (EBV, HHV-4), which is associated with Burkitt’s lymphoma. These vaccines are still under preclinical evaluation [80].

VLP vaccines to prevent HHV-8-related cancer, mainly exploiting several glycoproteins responsible for virus attachment to target cells, such as gpK8.1, gB, and gH/gL, have been shown to be effective in inducing a neutralizing antibody response [81,82,83].

Various strategies are now under preclinical evaluation to develop VLP vaccines to prevent EBV-related cancers [15]. The EBV genome does not belong to the standard vectors used due to manipulation difficulties. Thus, several studies have presented recombinant EBV genomes that express self-assembling features but are devoid of viral oncogenes, glycoproteins expressed on virion envelope (e.g., gp350, gp350/220, gB, and gH/gL) and essential for EBV entry, or EBV nuclear antigen 1 (EBNA1) and latent membrane protein 2 (LMP2) [84,85,86,87].

## 4. Target Antigens for Therapeutic Cancer Vaccines

The selection of an appropriate antigen is a crucial factor in determining the effectiveness of a vaccine when used in cancer treatments. An ideal antigen should be exclusively expressed in cancer cells, not found in healthy cells, be vital for the survival of the cells, and stimulate a strong immune response [4,88]. While targeting highly expressed molecules is a potential approach, it is important to take into account the potential for autoimmune responses towards normal tissues that express these molecules at lower levels, such as HER-2 in cardiac cells [89,90].

Neoantigens, which are novel proteins unique to tumor cells, are produced when somatic genomic alterations, including single-nucleotide variants (SNVs), base insertions and deletions (INDELs), and gene fusions, arise within the DNA of the tumor [91,92,93,94].

The tumor mutational burden (TMB) that leads to neoantigen formation is a major problem in several cancers (e.g., melanoma, lung adenocarcinoma, stomach adenocarcinoma, colorectal carcinoma, and sarcomas), mainly due to the consequent acquired resistance to therapies, especially immune checkpoint inhibitors [95,96,97]. On the contrary, it represents an advantage for immunotherapy because neoantigen-specific T cells are less likely to be eliminated during tumor immune evasion. Thus, the generation of a neoantigen can initiate a T-cell response specific to the tumor, thereby minimizing “off-target” effects, which can be harnessed for the development of personalized immunotherapy with cancer vaccines [91,98,99,100,101]. Some of the most extensively researched neoantigens for vaccines and immunotherapy are clonal neoantigens of KRAS, BRAF, and PIK3CA driver genes, but many others are under investigation as therapeutic vaccine targets in several cancers [94,100,102,103].

Various antigens, including oncoantigens, overexpressed antigens, cancer–testicular antigens (MAGE and NY-ESO-1), and foreign “nonself” antigens originating from viruses, also have potential in the development of cancer vaccines [88,104].

Several categorizations of tumor antigens have been proposed that highlight specific aspects to identify efficient targets. For example, oncoantigens are defined as persistent tumor antigens that have a causal role in tumor progression and do not escape from immune recognition [5,105,106]. They are typically expressed at low levels in normal cells but are overexpressed and/or amplified in tumor cells. Examples include EGFR, HER-2, the mucin MUC1, CD20, and the idiotypes of neoplastic clones of B and T cells [5,105,107]. Another example of a promising oncoantigen includes the insulin-like growth-factor-1 receptor (IGF1R), which is involved in the progression of epithelial and mesenchymal tumors [108,109]. Its co-targeting with other oncoantigens, such as HER-2, can induce a reduction in the invasive potential of cancer cells, including in trastuzumab-resistant cells [110,111], or those that cooperate in the establishment of the tumor microenvironment, such as vascular endothelial growth factor receptor (VEGFR) and platelet-derived growth factor receptor beta (PDGFR-β), which are also involved in the progression of several cancers, carcinomas, and sarcomas [112,113,114,115,116]. Oncoantigens can also be classified into three classes according to their location in cancer cells [6]. While class I antigens are located on the plasma membrane of tumor cells, class II antigens are not directly expressed by cancer cells but are present in the tumor microenvironment. Lastly, class III antigens are intracellular cancer cells antigens [117,118]. Crucially, not all tumor antigens can be considered oncoantigens due to their cellular location (e.g., MAGE) or because they are not drivers of tumor growth (e.g., carcinoembryonic antigen (CEA)) [106].

Tissue lineage and differentiation antigens, which are found in both normal and tumor cells originating from the same tissue, need to be targeted with care due to potential toxicity to some normal cells (e.g., B cells and prostate cells), leading to side effects but not mortality. Prostatic acid phosphatase (PAP), prostate-specific antigen (PSA), glycoprotein 100 (gp100), and melanoma antigen recognized by T cells 1 (MART-1) are among the most researched differentiation antigens [88,119].

Cancer germline antigens (CGAs), which are only expressed in germ cells of immune-privileged organs, epigenetically silenced in somatic tissues, and re-expressed in high levels in several carcinomas and sarcomas (such as MAGE and NY-ESO-1) [120,121], are another example of tumor-specific antigens targetable by cancer vaccines [122].

A newly identified group of targetable antigens involves molecules associated with epithelial-to-mesenchymal transition (EMT) and stemness, such as OCT-4, CD44, and CD133, as well as breast cancer stem cells, such as xCT [123,124]. A combined approach including EMT and/or stemness targets and target therapies might reduce tumor relapse and progression in several cancers, such as HER-2-positive breast cancer, in which cancer progression is often linked to acquired EMT and stemness features [116,124,125,126,127].

## 5. Prophylactic and Therapeutic VLP Cancer Vaccines for Breast Cancer

### 5.1. VLP Vaccines against HER-2

Several VLP-based vaccines have been developed and tested for the treatment of breast cancer, many of which present HER-2 as target [128]. HER-2 amplification and/or overexpression is observed in 20–30% of invasive breast carcinomas, and it is correlated with poor prognosis. Due to its overexpression only in cancer cells and its immunogenicity, HER-2 is an ideal target for the development of cancer immunotherapies.

A proof-of-concept study was performed on a prototypic HER-2-VLP vaccine, where the full extracellular domain (ECD) of HER-2 was attached on Acinetobacter phage 205 (AP205)-derived VLPs by a Tag/Catcher conjugation system [12,22,129]. The vaccine was effective in breaking immunological tolerance and induced higher antibody titers than a DNA-based vaccine in human HER-2 transgenic mouse models. Moreover, prophylactic vaccination with the HER-2-VLP vaccine reduced spontaneous development of mammary carcinomas by 50–100% in human HER-2 transgenic mice and inhibited the growth of HER-2-positive tumors implanted in wild-type syngeneic mice [129].

The HER-2-VLP vaccine was then re-engineered for human administration (referred to as ES2B-C001) by ExpreS^2^ion Biotechnologies (Hørsholm, Denmark) and showed promising results in HER-2-positive mammary carcinoma prevention and therapy in mice. In preclinical studies, the vaccine was administered alone, exploiting the intrinsic adjuvanticity of VLPs, or with Montanide ISA 51. The ES2B-C001 vaccine administered with adjuvant completely inhibited tumor growth in FVB mice challenged with a human HER-2-positive cell line (named QD), and mice remained tumor-free for more than one year after cell injection, whereas all control mice developed progressive tumor within 1–2 months. Moreover, 70% of mice treated with the vaccine without adjuvant were tumor-free. The vaccine (both with and without adjuvant) was also effective in completely inhibiting metastasis outgrowth in FVB mice challenged intravenously with QD cells; in contrast, control mice developed a mean of 300 lung nodules. We also evaluated the prevention and therapeutic efficacy of the vaccine in a tolerant mouse model transgenic for the HER-2 Delta16 isoform (Delta16 mice, FVB background [130]). Vaccine prevented spontaneous tumor onset in 95% of the Delta16-treated mice for more than one year (mice are still tumor-free at two years of age), while the control group developed progressive spontaneous tumors within 4–8 months of age. In this HER-2-tolerant mouse model, the vaccine confirmed its therapeutic efficacy on metastasis treatment by removing lung nodules in the treated mice. Additionally, the strong antitumoral activity was accompanied by a copious induction of anti-HER-2 antibodies of all IgG subclasses in both mouse models (ranging 1–10 mg/mL in FVB mice and 0.1–1 mg/mL in Delta16 mice) that remained stable for around one year after the last vaccination, suggesting the induction of a persistent B memory response [26].

Hu and Steinmetz developed a VLP-based vaccine against the HER-2-derived CH401 peptide epitope, containing epitopes for B cells and helper T cells, loaded on Physalis mottle virus (PhMV) VLPs. They compared two vaccine formulations, one loaded with a TLR-9 agonist (CpG-PhMV-CH401) and one with the antigen alone (PhMV-CH401), but no differences were observed between the two vaccine compositions, probably because the TLR-9 agonist loaded on the VLPs surface might have been dissociated in vivo. Both vaccines induced specific anti-CH401 IgG antibody response, although anti-PhMV antibodies were detected with a lower affinity relative to HER-2. Moreover, both sera of mice immunized with candidate vaccines and sera of mice that received the control VLPs bound HER-2-positive cells (DDHER-2), indicating a nonspecific binding of control-induced antibodies. Complement–dependent cytotoxicity (CDC) was confirmed only in mice vaccinated using vaccine candidates and not in control mice. These data were also reflected in the therapeutic efficacy of the vaccine in BALB/c mice preimmunized and then challenged with DDHER-2 cells: PhMV-CH401 improved survival in comparison to untreated mice (from 17 to 38 days) and PhMV-treated mice (from 24 to 38 days), but PhMV alone slightly prolonged mice survival (from 17 to 24 days) [131].

An interesting baculovirus-derived VLP platform exhibiting insect cells N-linked glycosylation composition (HER-2ic) or a mammalian-like N-linked glycosylation (HER-2ma) pattern was generated by Nika and colleagues and evaluated for the treatment of HER-2-positive mammary carcinoma. The antitumoral activity of the vaccines was evaluated in BALB/c mice pretreated with candidate vaccines and control VLPs with or without adjuvants (AddaVax or Poly (I:C)) and then challenged with the HER-2-positive mammary carcinoma cell line TuBo. While HER-2ma did not show antitumoral efficacy with or without adjuvant, HER-2ic had a better therapeutic activity alone or with AddaVax (20% of mice showed stable disease 70 days after cell challenge; in all the other groups, mice showed tumor progression within 40 days). The higher HER-2ic in vivo efficacy seemed to be linked to an effector function of the anti-HER-2 antibody response and the CD4^+^ and CD8^+^ T-cell response induced by the vaccine [132].

Several other anti-HER-2 cancer vaccines based on different VLP platforms have shown to be effective in inducing specific antibody effects and having anticancer activity (table under Section 7) [128,133,134,135,136,137].

Anti-HER-2 VLP-based vaccines can also be exploited for other tumors showing HER-2 expression, such as gastric cancer [138] and sarcomas [139].

### 5.2. Other Breast Cancer Antigens

A promising strategy for the treatment of high metastatic breast cancers, including HER-2-positive and triple-negative (TNBC) breast cancers, may be targeting breast cancer stem cells (BCSCs) [124,140].

The cystine–glutamate antiporter protein xCT was found to be highly expressed in BCSC and present at low levels in few types of normal cell lines [141,142]. Bolli et al. produced and tested a novel MS2 VLP cancer vaccine that displays the 6th ECD (ECD6 presents full homology in human and mouse sequence) of human xCT (AX09-0M6). AX09-0M6 vaccine stimulated a specific anti-xCT antibody response (not induced by MS2 VLP alone) in immunized mice. Moreover, purified IgG1 from vaccinated mice sera bound tumorspheres of murine (TuBo, 4T1) and human cell lines (MDA-MB-231, HCC-1806) and inhibited their formation and growth in vitro. To evaluate antitumoral activity, BALB/c mice were immunized twice with AX09-0M6 or MS2 VLPs and then challenged i.v. with TuBo-derived tumorspheres. While vaccination significantly reduced lung micrometastases, both treatment with control VLPs and the vaccine candidate altered immune infiltrates the lung microenvironment, increasing NK and CD8^+^ T cells. Thus, the intrinsic ability of MS2 VLPs to activate NK cell recall along with antigen-specific antibody response, mainly of the IgG2a subclasses, resulted in a significantly higher ADCC response related to AX09-0M6 administration. Furthermore, AX09-0M6 slightly reduced local tumor growth of 4T1 (HER-2-negative) cells and significantly reduced their spontaneous metastatic ability [123]. In a later study, the authors tested an analogous vaccine displaying the 3rd ECD of xCT [133]. These data encourage the targeting of BCSCs to prevent breast cancer progression, which might be a promising approach if combined with HER-2 target immunotherapy to overcome therapy resistance or tumor relapse driven by BCSCs.

IGF1R could be a good target to hinder breast cancer progression. A chimeric VLP vaccine based on the viral protein VP2 of human parvovirus B19 (B19V) and displaying two different IGF1R epitopes was developed and tested against the 4T1 cell line. Vaccine elicited specific anti-IGF1R antibody response in BALB/c mice and hampered in vivo tumor growth of 4T1 cells. Meanwhile, antibodies against the naïve vector V2 VLPs were elicited too, and the V2 VLP-vaccinated mice presented reduced or destroyed tumor growth [143].

IL-33 levels are associated with poor prognosis in several cancers due to its contribution to the development of immunosuppressive tumor microenvironment by affecting tumor stromal cells through the activation of carcinoma-associated fibroblasts (CAF) and the induction of VEGF expression [144,145]. Feng et al. developed HBcAg VLPs decorated with mature IL-33 and performed preventive and therapeutic studies on BALB/c mice challenged with 4T1 murine breast cancer cells. Both in preventive and therapeutic settings, the vaccine inhibited tumor growth and metastasis compared to control groups and induced specific antibody response. The mechanism of action of this vaccine may involve the induction of IFN-γ response along with a significant reduction of regulatory T (Treg) cells and myeloid-derived suppressor cells (MDSCs) into the tumors [146].

MUC1 is a good target in various adenocarcinomas, such as colon, breast, lung, and pancreatic cancers [147]. The prophylactic and therapeutic efficacy of VP2 B19-VLPs decorated with P53 and MUC1 epitopes (cellular and humoral epitopes, respectively) was evaluated in the 4T1 model. Both in prophylactic and therapeutic schemes, the TAA-VLP vaccine significantly inhibited tumor growth; however, a delay in tumor growth was also induced by the control WT-VLPs. Otherwise, while TAA-VLPs significantly reduced metastasis in both treatment regimens, the WT-VLPs did not affect metastasis outgrowth [148].

### 5.3. Targeting of Neoantigens through VLPs

VLPs can also be a feasible platform to develop vaccines against tumor-specific neoantigens. Mohsen and coworkers developed a VLP antitumor vaccine using short or long neoantigenic peptides of 4T1 mammary carcinoma cell line. VLPs were built with bacteriophage Qβ (Qβ-VLPs) and packaged with G10, a TRL-9 ligand, to enhance IFNα response. The vaccines tested in this work were composed of four VLPs, each presenting a different neoantigen (NeoAg) selected after proteomic and transcriptomic analysis. The authors identified four NeoAgs from the aggressive and low mutational burden mammary carcinoma cell line 4T1 and evaluated Qβ-NeoAg treatment efficacy against 4T1 cells in BALB/c mice. Both vaccines with short (Qβ-NeoAg_S_) and long (Qβ-NeoAg_L_) peptides significantly hindered tumor growth compared with the control, but Qβ-NeoAg_L_ showed an improved antitumor efficacy accompanied by increased CD8 and CD4 T cells and lower MDSC tumor infiltration. The authors also explored the differences between the immune response mediated by single and multitarget Qβ-NeoAg_L_ and observed that splenocytes of mice vaccinated with the multitarget vaccine induced significantly increased TNF-α production in CD8 and CD4 T cells and IFN-γ in CD8 but not CD4 T cells compared to mice vaccinated with a single NeoAg [149]. Thus, this study indicated that targeting patient-specific NeoAg might represent a therapeutic strategy for triple-negative high-grade breast cancer (in particular as combined therapy), and VLPs could constitute a feasible technology for personalized cancer vaccine [149,150].

## 6. VLP Cancer Vaccines in Melanoma Treatment

Melanoma is responsible for 80% of skin-cancer-related deaths worldwide and is characterized by the highest mutational burden of all human tumors [151,152].

In 2005, Brinkman et al. used polyomavirus-like particles (PLPs) as a vaccine platform in a murine melanoma model. The major coat protein VP1 was fused with a nonself antigen (ovalbumin, OVA257-264) or a self-antigen (tyrosinase-related protein, TRP2180–188) for the induction of CD8 T cells; both antigens were H2-K^b^-restricted T-cell epitopes. The therapeutic efficacy of VP1-OVA_252–270_ and VP1-TRP2_180–188_ was evaluated in MO5 (B16-OVA) melanoma-bearing C57BL/6 mice. VP1-OVA_252–270_ increased survival by 80–100%; in contrast, VP1-TRP2_180–188_ enhanced survival up to 60%. Furthermore, the authors showed that both vaccines were able to induce a CTL response [153].

A Qβ (G10)-Melan-A vaccine was shown to be cross-presented by dendritic cells, reaching the lymph nodes and generating T-cell responses in vitro and in HLA-A*0201 transgenic mice. Qβ-VLPs were packaged with the TLR-9 ligand G10 (a type-A CpG) as an adjuvant to increase stimulation of DCs and CD8 T cells and linked to the melanoma-differentiation-specific antigen Melan-A/Mart1. In 2010, the human VLP-based vaccine Qβ(G10)-Melan-A was tested in a phase I/II study in stage II/IV melanoma patients, inducing a T-cell-specific response in the majority of patients (63%), both in early and late clinical stages of metastatic melanoma [13,154]. A subsequent phase IIa clinical trial on Qβ(G10)-Melan-A vaccine formulated with different adjuvants (Montanide or topical imiquimod) was carried out. The majority of patients (76%) generated a specific T-cell response; furthermore, the vaccine candidate combined with imiquimod induced more central memory T cells than vaccine alone. The results showed 86% of treated patients exhibited a long-lasting immune response in draining lymph nodes for more than one year, with some patients presenting late-onset loss of Melan-A expression by tumor cells in situ, showing the outgrowth of antigen loss tumor variants and suggesting that multiple peptides might be necessary to fully eliminate tumors [13,155].

Several phase I and II clinical trials have been performed in melanoma patients treated with Qβ-VLPs loaded with A-type CpGs (G10) (GMP-001), which does not contain any tumor antigen, as monotherapy or in combination with anti-PD-1 check-point inhibitors, and the results showed promising clinical activity (NCT02680184, NCT03084640, and NCT03618641) [156,157,158].

To obtain a multiple-target and personalized vaccine most effective for the treatment of melanoma, Mohsen and colleagues successfully used a customized VLP platform loaded with TLR ligands. The authors developed three distinct sets of multitarget vaccines (MTV) specifically designed to combat the aggressive B16F10 murine melanoma. The first set of vaccines, known as germline epitope-based MTV (GL-MTV), was developed based on immunopeptidomics analysis, which identified key epitopes present in the germline. The second set, called Mutated-MTV, utilized epitopes predicted through whole-exome sequencing, specifically targeting mutated epitopes. Lastly, they created a combination set, Mix-MTV, integrating germline and mutated epitopes. Both GL-MTV and Mutated-MTV significantly hindered B16F10 tumor growth, but Mix-MTV showed a higher efficacy accompanied by an increase in CD8 T cells in the tumor [150]. Furthermore, Mohsen and coworkers implemented previously developed nanoparticles derived from cucumber mosaic virus (CuMVTT-VLPs) incorporating a universal tetanus toxoid epitope TT830–843 [159] by coupling p33. The CuMVTT-p33 nano-sized vaccine was then formulated with the micron-sized microcrystalline tyrosine (MCT) adjuvant, and its efficacy was compared to other adjuvants. CuMVTT-p33 added with MCT adjuvant increased the specific T-cell response in the B16F10p33 murine melanoma model and induced higher CD8 T-cell response and therapeutic efficacy compared to the Alum adjuvant [160].

MUC1 was also evaluated as a potential target in melanoma. A MUC1 VLP vaccine was built using Qβ phage and applied for the treatment of B16-MUC1 melanoma cells showing a significant reduction in lung nodules in immunized mice [161].

Another therapeutic approach explored was the design of a dual-antigen delivery system based on hepatitis B virus core antigen virus-like particles (HBc-VLPs) displaying OVA and/or gp100 peptides. On B16-OVA, the hybrid VLP vaccine had the best antitumor effect compared to the control and single-antigen-treated mice, both in local tumor and metastasis therapy [162].

Finally, a recent study conducted by Besson et al. reported the prophylactic and therapeutic efficacy of a VLP platform derived from human adenovirus type 3 (ADDomer) exploited to display MHC I and MHC II epitopes from OVA (ADD-Duo). Prophylactic vaccinations completely inhibited B16-OVA tumor growth for up to 17 days after the challenge, when control mice were all positive and the tumor reached the ethical volume limit in one of them. Vaccination also improved mouse survival in a therapeutic setting [163].

## 7. VLP Vaccines in Other Cancer Types

Pancreatic cancer has the worst survival rate among human tumors, and mortality is almost overlapping with incidence [164]. Two main targets were selected to treat pancreatic cancer models with VLP vaccines. The first one was murine TROP2 (mTROP2) loaded on the enveloped simian immunodeficiency virus (SIV)-VLP and tested in a syngeneic murine pancreatic cancer model (Panc02-mTrop2 cell line) (Table 1). Immunization of C57BL/6-tumor-bearing mice significantly reduced tumor growth and resulted in higher infiltration of CD4, CD8 T, and NK mTROP2 specific cells compared to the control groups. The addition of chemotherapy (gemcitabine) to the vaccine treatment further enhanced therapeutic efficacy and survival [165]. Another selected target was mesothelin (MSLN), which is overexpressed by a large number of pancreatic cancers [166,167]. Li and coworkers in 2008 studied the effects of MSLN overexpression in pancreatic cancer preclinical models and developed a prototype of SHIV VLP vaccine displaying human MSLN (hMSLN). The vaccine increased survival (9 weeks vs. 4 weeks after tumor cell challenge) and significantly reduced the tumor burden. Along with antitumoral activity, the hMSLN-VLP induced a specific MSLN antibody response and reduced Treg cells in the spleen and within tumors of treated mice [166]. A further study evaluated whether mouse MSLN (mMSLN) could break the immunological tolerance to mMSLN by exploiting VLPs. Similar to what was reported for hMSLN, mMSLN-VLP also reduced Panc02 cell growth in C57BL/6 syngeneic mice, induced CD8+-specific response, and reduced Treg cells [168]. Phase I/II clinical trials of GMP-001 vaccine in combination with INCAGN01949 (an activating anti-OX40 antibody) are under evaluation on stage IV pancreatic cancer (and other cancers except melanoma) patients, and in situ intratumoral injection led to tumor cell death (NCT04387071) [169,170].Many preclinical studies are ongoing to generate VLP-based vaccines to treat cervical cancer, and most of them target E6 and E7 oncoproteins. Monroy-Garcia et al. designed an HPV-16 L1 VLP fused with multiple E6 and E7 epitopes. When tested in C57BL/6 mice bearing TC-1 tumors, the vaccine produced a significant reduction (57%) in tumor size after three administrations. Moreover, the vaccine elicited persistent IgG1 antibodies for more than 1 year [171]. A VLP-E7 vaccine was produced by loading HPV-16 E745–98 on infectious bursal disease virus (IBDV) VLPs. This vaccine achieved a 100% survival rate in human HLA-A2 transgenic C57BL/6 mice with complete eradication of TC-1/A2 tumors [172]. A heterologous rabbit hemorrhagic disease virus (RHDV)-based VLP vaccine containing both Th epitope PADRE and CTL epitope HPV-16 E648–57 reduced TC-1 tumors in C57BL/6 mice by 50% and doubled survival time compared to the control group [173]. Qβ-VLPs packaged with the TLR-9 ligand G10 were also loaded with E7 protein from human HPV and evaluated as treatment in a preclinical cervical cancer model. The authors discovered that a subpopulation of resident DCs of draining LNs simultaneously took up both E7 and Qβ VLPs; thus, linking of Qβ and E7 was not required for uptake by the same DCs. Moreover, both E7 proteins mixed or coupled with Qβ-VLPs were effective in inducing strong CD8 and CD4 T-cell responses and significantly reducing tumor growth of TC-1 cells expressing HPV16 E7 oncoprotein in C57BL/6 female mice. However, a covalent linkage of E7 to CpG-loaded VLPs was required to elicit appropriate levels of antibody responses [174].

Some examples of therapeutic VLP cancer vaccines can be found against hepatocellular carcinoma (HCC). Hepatitis B virus core (HBc) particles were used as the carrier of single or multiple HCC epitopes: MAGE-1(278–286aa), MAGE-3(271–279aa), AFP1 (158–166aa), or AFP2 (542–550aa). DCs pulsed with the vaccine induced stronger CTL activity and greater IFN-γ secretion by responding T cells compared with peptide-pulsed DCs in HLA-A∗0201/k^b^ transgenic mice. The growth of established B16-pIR-HH tumors was significantly inhibited by immunization using VLP-pulsed DCs, resulting in a higher survival rate of vaccinated mice [175]. A further multiepitope VLP vaccine was designed by Ding et al. by loading HBc VLP of four HBx-dominant CTL epitopes (HBx_(115–123)_, HBx_(92–100)_, HBx_(140–148)_, or HBx_(52–60)_). VLP-pulsed dendritic cells in both HLA-A*0201 transgenic (Tg) mice and peripheral blood lymphocytes from HLA-A2(+)/HBx(+) HBV-infected hepatocellular carcinoma (HCC) patients showed CTL responses against epitope-loaded VLPs. Along with higher immunogenicity, multiepitope VLPs demonstrated an enhanced antitumor activity compared to single-epitope VLP vaccines [176].

A multitarget chimeric VLP vaccine was proposed as a therapeutic strategy in a preclinical model of colorectal cancer. RHDV VP60 capsid proteins containing recombinantly inserted epitopes from murine topoisomerase IIα (T.VP60) and survivin (S.VP60) were tested as mono- or multi (TS.VP60)-target vaccines. Overall survival was significantly improved amongst C57BL/6 mice bearing MC38-OVA tumors and immunized with T.VP60 (60%), S.VP60 (60%), or TS.VP60 (73%). Cured mice were then rechallenged with MC38-OVA cells and none of them developed tumors [177].

A chimeric bovine papillomavirus (BPV) VLPs displaying MUC1 induced MUC1-specific CTL in a human MUC1 transgenic mouse model and impaired, or eradicated in few mice, tumor growth induced by RMA-MUC1 cells (a T cell lymphoma line) in this mouse model [178].
ijms-24-12963-t001_Table 1Table 1VLP therapeutic vaccines: preclinical studies.Cancer TypeCell LineMouse ModelTumor AntigenVLP PlatformAdjuvant or Combination TherapyType of StudyReferencesBreast cancer MamBo89 (HER2-positive cell line derived from a hHER2 transgenic mouse model)FVB (FVB/NCrl) F1 HER2/Delta16 (FVB background)HER2AP205 phageNoneProphylactic and therapeutic[129] Breast cancerD16-BO-QD (HER2-positive cell line derived from a hHER2 transgenic mouse model)FVBDelta16 (FVB background)HER2AP205 phageMontanide ISA 51Prophylactic and therapeutic[26]Breast cancerDDHER2 (mouse cell line expressing rat HER2)BALB/cCH401 (rat HER2-derived epitope)Physalis mottle virus (PhMV)CpG (TLR-9 agonist loaded on VLPs)Prophylactic and therapeutic[131]Breast cancerTuBo (HER2-positive cell line derived from a NeuT transgenic mouse model)BALB/cHER2Recombinant baculovirus (rBV)Glycosylation patterns AddaVax Poly (I:C)Prophylactic[132]Breast cancerTuBoBALB/cGP2 (HER2/neu derived peptide)Bacteriophage lambda (λF7)NoneProphylactic and therapeutic[134]Breast cancerTuBoBALB/cE75 (HER2-derived peptide)λF7NoneProphylactic and therapeutic[135]Breast CancerD2F2/E2 (mouse cell line transfected with hHER2)BALB/cGPI-HER2rBVNoneProphylactic[136]Breast cancerTuBo4T1BALB/cxCTMS2NoneProphylactic and therapeutic[123,133]Breast cancer4T1BALB/cIL-33HBcAgNoneProphylactic and therapeutic[146]Breast cancer4T1BALB/cP53 and MUC1VP2 B19NoneProphylactic and therapeutic[148]Breast cancer4T1BALB/cNeoAGQβG10 (TLR-9 agonist loaded on VLPs)Prophylactic and therapeutic[149]MelanomaMO5 (B16-OVA)C57BL/6OVA_257–265_ TRP2_180–188_VP1 PLPsNoneTherapeutic [153]MelanomaN/AHLA-A*201 TgMelan-AQβG10 N/A[154]MelanomaB16F10C57BL/6 C57BL/6Rag2^−/−^Germiline and mutated epitopesQβB-type CpGsTherapeutic [150]MelanomaB16F10p33C57BL/6 C57BL/6Rag2^−/−^p33CuMVTMicrocrystalline tyrosine (MCT)Therapeutic [160]MelanomaB16-MUC1MUC1.Tg (C57BL/6 background)MUC1QβMonophosphoryl-Lipid A (MPLA)Prophylactic[161]MelanomaB16-OVAC57BL/6OVA and gp100HBcNoneTherapeutic [162]MelanomaB16-OVAC57BL/6MHC I and II OVA epitopesHuman adenovirus type 3 (HAdV)ODN 2395MPLApoly(I:C)Prophylactic and therapeutic[163]Pancreatic CancerPanc02-mTrop2C57BL/6Trop2SIVGemtabicineTherapeutic [165]Pancreatic CancerPanc02C57BL/6hMSLNSHIVNoneTherapeutic [166]Pancreatic CancerPanc02C57BL/6mMSLNSHIVNoneTherapeutic [168]Cervical cancerTC-1C57BL/6E6 and E7HPV-16 L1None Prophylactic and therapeutic[171]Cervical cancerTC-1/A2C57BL/6E7IBDVNone Therapeutic [172]Cervical cancerTC-1/A2C57BL/6E6RDHVNoneTherapeutic [173]Cervical cancerTC-1C57BL/6E7QβG10 Therapeutic [174]Hepatocellular carcinomaB16-pIR-HHHLA-A*0201/k^b^ Tg (C57BL/6 background)MAGE-1MAGE-3AFP1AFP2HBcNoneTherapeutic [175]Hepatocellular carcinomaEL-4HLA-A*0201 TgHBx_(115–123)_ HBx_(92–100)_ HBx_(140–148)_ HBx_(52–60)_
HBcNoneProphylactic[176]Colorectal cancerMC38-OVAC57BL/6Topoisomerase IIα and survivinRHDV VP60CpGsTherapeutic [177]T cell lymphomaRMA-MUC1MUC1 Tg (C57BL/6 backgroung)MUC1BPVNoneProphylactic[178]N/A, not available.

## 8. Conclusions

Cancer vaccines represent a broad immunological antitumoral approach due to their ability to stimulate both innate and acquired immunity and induce an immune memory, leading to long-lasting protection. VLPs are an attractive technology to develop prophylactic and therapeutic cancer vaccines owing to the possibility of displaying individual or multiple epitopes with an intrinsic adjuvant activity. Preventive vaccines against HBV and HPV infections are already approved and successfully being used. In contrast, there are no approved therapeutic VLP cancer vaccines, but promising preclinical and early clinical data, mainly in HER-2-positive breast cancer and melanoma treatment, give hope that the goal could be reached.

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
