# Peer review of "Virus-like Particle (VLP) Vaccines for Cancer Immunotherapy"

_ijms, 2023, doi:10.3390/ijms241612963_

Round 1

Reviewer 1 Report

In the current manuscript, Ruzzi et al., comprehensively summarized the progress of Virus-like particles (VLPs) in cancer therapy. They provided detailed information about target antigen selection, VLP vaccine production and immune response after vaccine administration. The authors focus on the progress of VLP on HER2+ breast cancer, melanoma, prostate cancer et al.. They provided updated progress on preclinic and clinic therapies of VLP on these cancers.

In general, this paper is presented in a well organized and logistic way. The information is up to date and knowledgeable. I don't have any major questions.

Only 1 minor point, could the authors divide the summarized Figure into different parts and insert them to their desired position. This will be greatly helpful for the audience to follow.

Author Response

Ruzzi et al.

“Virus-like particle (VLP) vaccines for cancer immunotherapy”

International Journal of Molecular Sciences-2561812

Response to Reviewer 1

“In the current manuscript, Ruzzi et al., comprehensively summarized the progress of Virus-like particles (VLPs) in cancer therapy. They provided detailed information about target antigen selection, VLP vaccine production and immune response after vaccine administration. The authors focus on the progress of VLP on HER2+ breast cancer, melanoma, prostate cancer et al.. They provided updated progress on preclinic and clinic therapies of VLP on these cancers.

In general, this paper is presented in a well organized and logistic way. The information is up to date and knowledgeable. I don't have any major questions.

Only 1 minor point, could the authors divide the summarized Figure into different parts and insert them to their desired position. This will be greatly helpful for the audience to follow.”

Thank you for reviewing our manuscript and for your kind comments.

The Figure was designed and uploaded on the submission form as graphical abstract in order to summarize the article and give an overview of the review content before the reading. For this reason, authors would preserve it as graphical abstract instead of figures inserted into the text.

Reviewer 2 Report

Overall the paper is a nice review.

My only comments are regarding the discussion of safety:

On line 47 the authors say that  “They (VLPs) are intrinsically safe”, but that is only in regard of unintentional viral gene delivery. This should be specifically added to this sentence.

Regarding the overall safety of VLPs as vaccines:

It is well known that viral capsids of even defective or empty viral particles can trigger various innate immune responses. Their safety is always investigated on a case by case manner. Viral capsids can by specifically or non-specifically and internalized by various cells via varying mechanisms. Some of these events my trigger various cytokine release which may lead to inflammation. There should be no automatic presumption that de novo assembled nanoparticle sized VLPs containing various endogenous antigens are "inherently safe". There should be a section addressing these concerns and whatever safety data are available regarding vaccine VLPs, they should be referenced there.

Author Response

Ruzzi et al.

“Virus-like particle (VLP) vaccines for cancer immunotherapy”

International Journal of Molecular Sciences-2561812

Response to Reviewer 2

“Overall the paper is a nice review.

My only comments are regarding the discussion of safety:  On line 47 the authors say that “They (VLPs) are intrinsically safe”, but that is only in regard of unintentional viral gene delivery. This should be specifically added to this sentence.

Regarding the overall safety of VLPs as vaccines:

It is well known that viral capsids of even defective or empty viral particles can trigger various innate immune responses. Their safety is always investigated on a case by case manner. Viral capsids can by specifically or non-specifically and internalized by various cells via varying mechanisms. Some of these events my trigger various cytokine release which may lead to inflammation. There should be no automatic presumption that de novo assembled nanoparticle sized VLPs containing various endogenous antigens are "inherently safe". There should be a section addressing these concerns and whatever safety data are available regarding vaccine VLPs, they should be referenced there.”

Thank you for reviewing our manuscript and for your suggestions.

The sentence in the introduction on VLPs safety was revised in “Thanks to the lack of genetic material needed for viral replication, VLPs inherently ensure safety against unintentional viral gene delivery” (lines 45-46).

A paragraph on possible toxicity of VLPs-based vaccines was added in “Virus-like particles (VLPs): an overview” section (lines 125-135):

“[…] It is crucial to evaluate any possible side effects which might be related to auto-immunity caused by the antigen or allergic reactions and excessive immune responses against VLPs (14, 44). Since VLP vaccines are designed to induce both humoral and cellular response, each component (coat protein, chemical cross-linker, the displayed antigen and adjuvant) could present potential toxicity (14). Some examples of adverse effects are the pre-existing immunity against vaccine carrier proteins, the presence of unconjugated self-antigen that may bind healthy cells or an excessive immune response induced by the vaccine combined with an adjuvant (14, 45–48). Thus, for every new vaccine composition and administration schedule it is essential to study possible toxicity. Current data from approved VLP vaccines showed that they are generally safe (49)”

Round 2

Reviewer 2 Report

The revision addressed satisfactorily the concerns with the first variant of the manuscript. Thank you.